# Effect of aging and *Varroa* parasitism on the paracellular and transcellular permeability of the honeybee blood-brain barrier

Tyler Quigley[1]*, Gro Amdam[1,2]

1  School of Life Sciences, Arizona State University, Tempe, Arizona, United States of America, 2  Faculty of Environmental Sciences and Natural Resource Management, Norwegian University of Life Sciences, Aas, Norway

* tpquigle@asu.edu

## Abstract

Honeybees (*Apis mellifera*) provide crucial pollination services to agricultural systems globally, however, their healthspan in these contexts is constantly at risk. Agricultural environments impose a variety of sublethal stressors onto honeybees, including parasites, pathogens, pesticides, and poor nutrition. Synergies between age, age-associated tasks, and these stressors are believed to underlie colony failure trends of the past decade. Identifying the mechanisms by which age and stressors impact honeybee physiology is an important priority in protecting honeybee and other pollinator populations. An underexplored physiological structure in honeybees is the blood-brain barrier, a protective layer of cells that surrounds the brain. Here, we assess key dimensions of blood-brain barrier function; paracellular and transcellular permeability to molecules in the hemolymph. We measured these modes of permeability across worker groups that differ in age and foraging experience, as well as in bees exposed to varying levels of infestation by the parasitic mite *Varroa destructor* during development. Our results demonstrate that the paracellular permeability of the honeybee blood-brain barrier is stable across these age groups and upon *Varroa* exposure. In contrast, we found that transcellular permeability is increased in honeybees exposed to a high *Varroa* load. Together, these results demonstrate that age-related variation and parasitic stress differentially impact a primary protective structure of the honeybee central nervous system, which may lead to targeted interventions for protecting honeybee healthspan. The assay developed here may be easily applied to different aging- and stress-related contexts, further enabling studies focused on understanding maintenance and decline of the honeybee blood-brain barrier.

## Introduction

Over the past half century, agricultural intensification has become the primary driver of insect population decline worldwide [1] Paradoxically, modern agricultural practices

**Data availability statement:** The data underlying the results presented in the study are available from Figshare, DOI: 10.6084/m9.figshare.30888278.

**Funding:** Research Council of Norway, Grant # 335244.

**Competing interests:** The authors have declared that no competing interests exist.

impose a variety of risk factors onto wild and managed bee pollinators, which are estimated to contribute hundreds of billions of USD to global crop production [2–4]. These risk factors include pests, pathogens, pesticides, climate stress, poor nutrition, and poor management practices, each of which has distinct consequences for pollinator health [5]. Pollinator health can be defined as "a state that allows individuals to live longer and/or reproduce more, even in the presence of pathogens, thus providing more ecological services" [6]. Efforts to improve pollinator health are thus mutually beneficial between our species and the insect species we rely so heavily upon. Fortunately, remediating and protecting pollinator populations has become a priority for a variety of stakeholders, and recent policy reports have identified key priority areas to improve insect pollinator health specifically [2,7,8]. Among these priorities is research to increase our understanding of the fundamental biological mechanisms by which stressors impact pollinator insect health.

Honeybees (*Apis mellifera*) are the most commonly managed insect pollinator species worldwide, and many crop production systems depend specifically on honeybee pollination [9]. Nevertheless, annual honeybee colony losses have remained high across Europe, Canada, and the United States for over a decade [10]. A "colony loss" can manifest in multiple ways, but the decline in colony performance over time leading up to a loss is a result of the collective decline in the health and performance of colony members, i.e., their ability to perform the tasks essential to colony survival across their lifespan [5,11–13].

Understanding how risk factors impact honeybee healthspan can provide a path forward for improving individual and colony health. Although a risk factor may affect multiple physiological systems at once, a honeybee's performance of a task is mediated by the central nervous system (CNS). Indeed, CNS dysfunction and decline seem to be a primary link between environmental stress, reduced individual healthspan, and colony failure [11,14]. The neuroanatomy and cognitive architecture of the honeybee CNS is well understood as honeybees are a well-established neurobiological model useful for probing the fundamental principles that govern animal behavior, brain health, and neurological disease and aging [15,16]. By leveraging and expanding this toolkit, we can better resolve the mechanisms by which risk factors dysregulate the CNS of honeybees over their life course, and contribute to reduced lifetime performance at individual and colony levels.

An underexplored region of the honeybee brain which may mediate the impact of stress on the honeybee CNS is the blood-brain barrier. Although arthropods contain hemolymph rather than blood, the term "blood-brain barrier" is also used to refer to the selective cellular barrier that regulates the access of molecules and signals from peripheral tissues to the brain [17,18]. Its primary function is to maintain the strict ionic environment necessary for optimal neuronal signaling, however, it is also enriched in a variety of active and facilitative transport systems responsible for the exchange of ions, nutritious metabolites, hormones, xenobiotics, drugs, and metabolic waste products between the brain and hemolymph [19]. The blood-brain barrier also acts as a homeostatic sensor which transduces information about the physiological state of an animal towards neuronal circuits [20,21]. Despite differences in the

cellular makeup and broad structure, many of the molecular systems that form and govern the blood-brain barrier are conserved between mammalian and insect blood-brain barriers [22,23]. The homology of these systems provides a uniquely robust basis for applying comparative insights across model species, and has been particularly useful in understanding the mechanisms of aging-related decline in blood-brain barrier function [24,25].

A key dimension of blood-brain barrier integrity is its permeability to molecules in circulation [26]. In many other species, the permeability of the blood-brain barrier is dynamic across an individual's lifespan and can be impacted by factors including nutrition, environmental stress, disease, and aging [27–29]. Although an animal's behavior is coordinated by neurons, neuronal function can be severely impaired by a leaky blood-brain barrier [17]. The consequences of increased blood-brain barrier permeability include increased oxidative stress, neurodegeneration, and cognitive decline [30,31]. These physiological deficits are also observed in honeybees upon increasing age and exposure to environmental stress. It is plausible to suspect then that effects of age and stress on honeybee healthspan are mediated by changes in blood-brain barrier permeability. Understanding the mechanisms by which these effects impact honeybee healthspan can inform the development of novel interventions to improve honeybee health.

Here, we present an adapted insect blood-brain barrier permeability assay to measure the transcellular and paracellular permeability in honeybees [32–34]. We applied this assay to honeybee workers to assess how blood-brain barrier permeability changes with age and how it is impacted by *Varroa destructor* parasitism. The assay described herein is simple and low-tech, which lends itself to adaptation for assessing insect blood-brain barrier permeability in diverse aging and stress contexts, thereby presenting new paths to understanding these impacts on a structure crucial for brain homeostasis.

## Methods

### Animals

Honeybees for this study were sourced from multiple distinct colonies maintained at the Arizona State University Bee Lab in Mesa, Arizona. Collections occurred between Fall 2020 and Spring 2022.

### Sample collection

**Aging.** Honeybee worker lifespan is plastic, regulated by a variety of intrinsic and extrinsic factors. Aging is task-dependent rather than chronologically dependent, and physiological aging markers can be reliably detected in honeybee workers ~14 days after their first foraging flight, irrespective of the worker's chronological age [35]. To assess how honeybee blood-brain barrier permeability changes with age, we collected nurse bees (pre-foraging), young foragers (<14 days of foraging), and old foragers (>14 days of foraging). To collect these age groups, a brood frame was removed from a colony and placed in a wire cage in an incubator overnight at 33°C and 70% humidity. The next day, approximately 250 newly emerged bees were obtained from this frame, marked on the abdomen, and placed back into the hive (nurse bees). On the same day as the newly emerged bees were placed in the hive, approximately 1,500 foragers were marked with a different color upon their return from foraging (old foragers). Fourteen days later, another subset of 200 foragers was marked upon their return from foraging (young foragers). The day after young foragers were marked, five bees of each group were collected and brought into the lab for dye permeability assays.

**Varroa.** To assess how infection with *Varroa* impacts honeybee blood-brain barrier permeability, we collected recently emerged honeybees exposed to 0–1 mites and 2–4 mites during their development. Colonies with *Varroa* infestation were identified and left untreated for the duration of sampling each Fall or Spring season. To collect newly emerged honeybees, brood frames with actively emerging brood were collected and observed indoors under low light. As bees emerged, they were collected with soft forceps. Adult mites present on the honeybee and within its cell were counted and removed. Honeybees were paint marked with a color corresponding to the number of mites found on the bee and in the cell and placed into a wire mesh cylinder. Sampling occurred over multiple days with ~10 total bees collected per

session. After each sampling session, the single wire mesh cage was placed into a host colony with no *Varroa* infestation. The specimens aged one day in the host colony to eliminate confounding development factors that may influence blood-brain barrier permeability. After 24 hours, the wire mesh cage was removed and brought into the lab for dye permeability assays.

## Dye permeability assay

We assessed paracellular and transcellular blood-brain barrier permeability in honeybees using an assay adapted from previous studies with honeybees and *Drosophila* [32,36]. These modes of permeability are assessed with dyes that have distinct interactions with the blood-brain barrier. Paracellular permeability refers to the movement of molecules through the intercellular space between cells. Paracellular integrity is maintained at the insect blood-brain barrier by septate junctions between blood-brain barrier cells, minimizing the intercellular space through which molecules can pass to enter the brain [19]. We used a Texas Red-conjugated 10 kDa dextran (TRD) (Invitrogen D-1863) to probe paracellular permeability, as this is the smallest molecular weight which is prevented from passing through the insect blood-brain barrier under healthy conditions [32,37].Transcellular permeability refers to the movement of molecules across the lipid bilayers of the cells that make up the blood-brain barrier [19]. One way in which the blood-brain barrier regulates this pathway is by maintaining a high concentration of ATP-binding cassette (ABC) transporters on the apical membrane of blood-brain barrier cells [38]. We used the dye Rhodamine B (Rho B) (Sigma R6626) to probe transcellular permeability, as it is permeable to cell membranes and is a substrate for the ubiquitous ABC transporter p-glycoprotein (P-gp), and possibly others [36].

To perform the assay, specimens were secured into a custom mount and placed under a dissecting microscope. Using a surgical blade, the hair on the left side of the head was shaved down and a sliver of cuticle was removed from the head, exposing the tissue inside. Effort was made to minimize the size of the cut and the handling of the removed cuticle so as to minimize disturbance to the underlying tissue. Often a layer of connective tissue remained intact immediately inside the incision, in which case the tissue was gently torn to enable dye permeation into the head capsule.

Either 1 μL of 0.125 mg/mL of RhoB dye or 1 μL of 5 mM 10-kDa dextran dye was then injected into the head capsule using a bevel-tip syringe (Lab Depot, 002105). Once injected, the bees were kept in the mounts, and placed onto a wet paper towel and covered with a cardboard box to reduce desiccation at the incision. After 45 minutes of exposure to the dye, bees were decapitated, and heads placed into a 1% paraformaldehyde solution. Brains were dissected under fixative, cleaned of surrounding tissue, and placed into a well in a 96 well plate. Wells were filled with 50 μL 0.1% SDS solution. SDS breaks up cell walls, releasing dye. Each brain was further processed within the well by repeated pressing with a pestle for 30 seconds. Once all brains were dissected, placed, and processed in the well plate, the dye concentrations present in each well of the well plate were analyzed on a BioTek HT1 well plate reader (Excitation: 535 Emission: 595).

## Analysis

The relative fluorescence values derived from the brains of each age group and each mite load group were used to compare paracellular and transcellular permeability between the groups. All analyses were performed in R (version 4.4.0).

Age group comparisons were made using ANOVA tests, followed by statistical power analyses to evaluate the sensitivity of the comparisons. Using the *pwr* package in R (version 4.4.0), prospective one-way ANOVA models were parameterized with the observed group sizes and effect sizes expressed as Cohen's *f* (derived from $\eta^2$). For each analysis, achieved power at $\alpha = 0.05$ and the minimal detectable effect size (MDE) corresponding to 80% power were estimated.

The comparison of paracellular permeability between mite load groups was made using a Wilcoxon rank-sum test. Effect size was expressed as rank-biserial correlation (r) with 95% confidence interval and converted to an approximate Cohen's d for power estimation (pwr.t2n.test, $\alpha = 0.05$, two-tailed). The comparison of transcellular permeability between mite load groups was made using a Welch's t-test. Effect size was calculated as Hedges' *g* with 95% confidence intervals, and power analyses were conducted using the *pwr* package in R to estimate achieved power and the minimal detectable

effect size at 80% power (*n* = 19 vs 21). Data are presented as boxplots depicting the median (bold center line), interquartile ranges (span of box), range within 1.5 times the interquartile range (whiskers), and outliers.

## Results

### Paracellular permeability

We did not find a significance difference in dextran dye concentrations in the brains of nurses, young foragers, and old foragers (ANOVA, $F_{2,22}$ = 1.73, p = 0.20) (Fig 1A, Table 1). The observed effect size was η² = 0.14 (ω² = 0.06; f = 0.40). Power analysis indicated that, with n = 8–9 per group, the test had only 36% power to detect this effect and would require an effect of f ≥ 0.67 for 80% power. These results suggest the experiment was underpowered to detect small or moderate age effects.

Similarly, we did not find a significant difference in brain dextran concentration between honeybee workers who emerged with a low mite load versus a high mite load (Wilcoxon, W = 141, p = 0.61) (Fig 1B, Table 1). The observed effect size was r = 0.11 (95% CI [–0.30, 0.49]), corresponding to AUC = 0.56. This equates to a very small difference (d ≈ 0.10), for which the design had only ≈ 6% power. Achieving 80% power would have required a large effect (d ≈ 1.06, r ≈ 0.53).

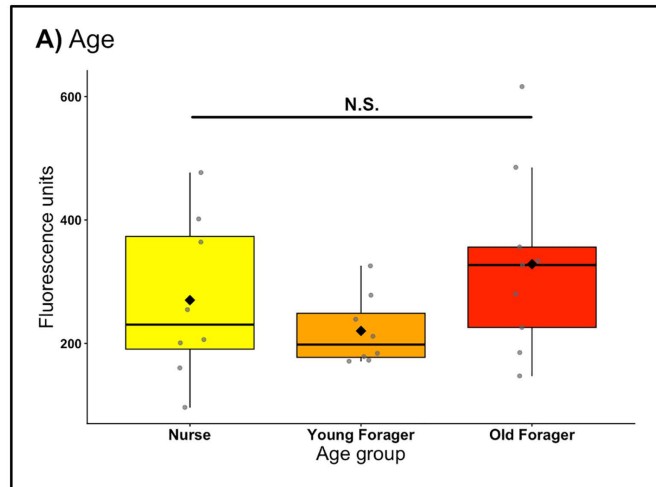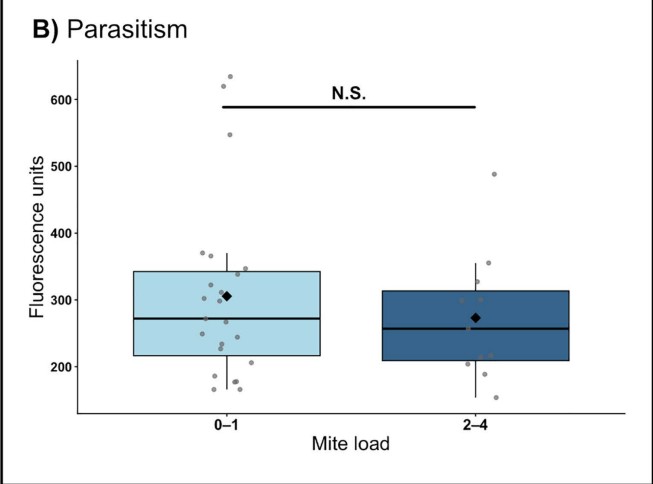

**Fig 1. Paracellular permeability of the honeybee blood-brain barrier.** There was no significant difference detected (A) among age groups (ANOVA, $F_{2,22}$ = 1.73, p = 0.20) nor (B) between bees emerging with low (0–1 mites) or high (2–4 mites) Varroa loads (Wilcoxon rank-sum test, *W = 141*, p = 0.61). Each point represents an individual bee; boxplots show the interquartile range with median line and overlaid mean (black diamond). **N.**S. = nonsignificant.

**Table 1. Summary statistics for paracellular permeability data.**

|  | n | Mean fluorescence | df | Test Stat | *p* |
|---|---|---|---|---|---|
| **Age** |  |  | **ANOVA** |  |  |
| Nurse | 8 | 270.1 | 2 | F = 1.73 | 0.20 |
| Young Forager | 8 | 220.3 |  |  |  |
| Old Forager | 9 | 328.4 |  |  |  |
| **Parasitism** |  |  | **Wilcoxon** |  |  |
| 0-1 mite | 22 | 305.5 | 0.812 | W = 141 | 0.61 |
| 2-4 mite | 8 | 273.1 |  |  |  |

## Transcellular permeability

Mean Rho B concentration in the brains of nurses, young foragers, and old foragers trended up with age, however, differences were not significant (ANOVA, $F_{2,27}=1.656$, $p=0.21$) (Fig 2A, Table 2). The observed effect size was $\eta^2=0.11$ ($\omega^2=0.04$; $f=0.35$). Although the model assumptions were met, the analysis had only 35% power to detect an effect of this magnitude and would require a large effect ($f\geq0.60$) for 80% power. Thus, subtle age-related differences may have gone undetected.

The brain Rho B concentration in honeybees that emerged with a high mite load was significantly higher than found in the brains of honeybees that emerged with a low mite load ($t(37.9)=-3.00$, $p=0.0047***$) (Fig 2B, Table 2). The effect size was large (Hedges' $g=0.95$, 95% CI [0.29, 1.60]), and the test achieved 83% power at $\alpha=0.05$. A sensitivity analysis indicated that a difference of $d\approx0.91$ would be detectable with 80% power under the present sample sizes ($n=19$, 21).

## Discussion

The healthspan and performance of a honeybee worker is closely linked to their CNS health [14,39]. The CNS's primary line of defense against potentially harmful endogenous and exogenous factors is the blood-brain barrier. Here, we

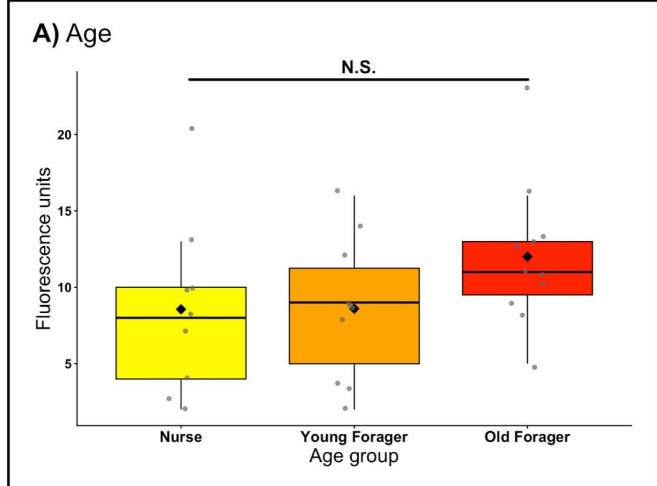
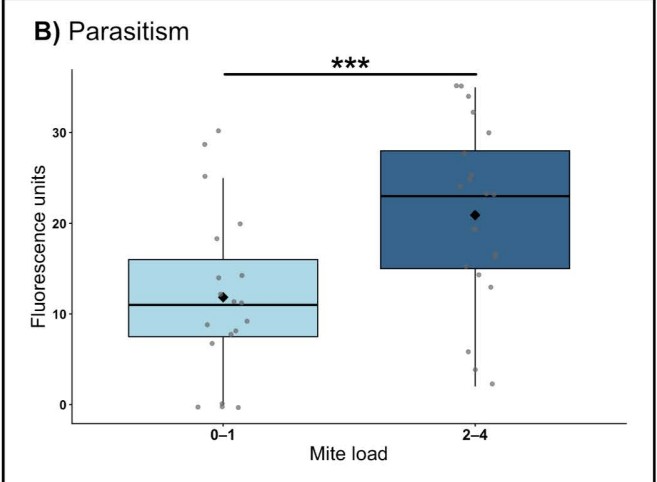

**Fig 2. Transcellular permeability of the honeybee blood-brain barrier. (A)** There was no significant difference in Rho B signal between age groups (ANOVA, $F_{2,27}=1.66$, $p=0.21$). **(B)** The Rho B signal observed from the brains of workers which emerged that two or more mites is significantly higher than the signal observed from the brains of workers which emerged with zero or one mite ($t(37.9)=-3.00$, $p=0.00473$). Each point represents an individual bee; boxplots show the interquartile range with median line and overlaid mean (black diamond). *** = $p<0.005$, **N.S.** = nonsignificant.

**Table 2. Summary statistics for transcellular permeability data.**

|  | n | Mean fluorescence | df | F | *p* |
|---|---|---|---|---|---|
| **Aging** |  |  | **ANOVA** |  |  |
| Nurse | 9 | 8.56 | 2 | 1.66 | 0.210 |
| Young Forager | 10 | 8.60 |  |  |  |
| Old Forager | 11 | 12 |  |  |  |
| **Parasitism** |  |  | **t-test** |  |  |
| 0-1 mite | 19 | 11.8 | −3.00 | 37.9 | 0.00473*** |
| 2-4 mite | 21 | 20.9 |  |  |  |

compared the paracellular and transcellular permeability of the honeybee blood-brain barrier across different worker age groups and between workers with low and high mite loads. As honeybee workers developed under natural colony conditions, our age groups represent ranges of worker experience and task exposure rather than precisely defined chronological ages. We did not find a difference in paracellular permeability across age groups or between mite load levels (Fig 1). Conversely, we found a non-significant increase in transcellular permeability in workers with age (Fig 2A), and significantly higher transcellular permeability in workers that emerged with a high mite load versus those that emerged with a low mite load (Fig 2B).

The insect blood-brain barrier is formed by the neural lamella and two layers of distinct glial subtypes [40,41]. The neural lamella is an acellular fibrous matrix that sheaths the entire nervous system. Perineurial glia (PG) compose the most outer cellular layer and subperineurial glia (SPG) form the layer immediately beneath PG. Paracellular diffusion is primarily blocked by a dense network of junctional protein complexes that bind the maze-like interface between SPG cells [42,43]. We assessed the viability of the paracellular barrier with a 10 kDa dextran dye, which is normally excluded by the SPG layer, but can permeate if the paracellular diffusion barrier is disrupted [32]. This assay is often used to assess the impact of genetic disruptions to the *Drosophila* BBB, however, it has also been employed to demonstrate how paracellular integrity is disrupted in a model of traumatic brain injury and parasite-induced summiting behavior [33,44,45].

We hypothesized that increasing age and exposure to *Varroa* load would disrupt the paracellular integrity of the honeybee blood-brain barrier. Our results do not support this hypothesis, and instead suggest that paracellular integrity is robust to aging and parasitic stress (Fig 1). The specimens collected for the *Varroa* permeability study were newly emerged, and the permeability assay was applied when they were one day old. Thus, this group also represents a younger age group than those assessed in the aging experiment. The amount of dye measured in the brains of specimens between the two experiments was similar, further suggesting that paracellular permeability is stable across lifespan and its quantification with this method repeatable across experiments. It is possible that paracellular integrity is stable over the lifespan of a honeybee and in response to stress due to the complex organization of junctional proteins within the intercellular space of SPG, though further work is needed to characterize the proteins that compose this space in honeybees [40].

The other path a molecule can take across the blood-brain barrier is diffusion across SPG cell membranes. This transit pathway is regulated by a number of chemoprotective mechanisms including ABC transport systems that shuttle molecules out of the barrier cells and into the hemolymph [18]. Rho B is a small molecule dye which can diffuse passively into the blood-brain barrier layer and is a substrate for P-gp and possibly other MDR transporters which constitute a major portion of blood-brain barrier efflux systems [36]. The concentration of brain Rho B is thus a measure of the efficiency of efflux activity at the barrier, which can be impacted by various stressors in honeybees [46]. We hypothesized that aging and *Varroa* load would decrease efflux activity, resulting in more dye accumulation in the brains of workers of increased age and in workers with higher mite load.

We did not find a significant difference in transcellular permeability between the age groups we examined (Fig 2A). However, our results show a trend towards an increased transcellular permeability with age and deserve a follow-up study with an increased sample size to determine whether this trend is a true effect. Our sampling strategy was based on collecting from behavioral categories that typically differ in age and foraging exposure, thus the oldest workers examined may not fully represent the most senescent individuals in the colony. Increases in blood-brain barrier transcellular permeability are observed with aging across multiple animal species including humans [25,47,48]. A looming question in blood-brain barrier aging research is whether alterations in blood-brain barrier transport systems reflect a failure of function or an adaptation to the changing needs of an aging brain [48]. The oldest honeybee workers in a colony are foragers, who engage in the collection of resources from the environment [49]. It is possible that blood-brain barrier efflux mechanisms are downregulated during this life stage to allow for the increased uptake of endogenous molecules beneficial to the brain in a foraging context. In the modern agricultural setting, however, a leaky blood-brain barrier leads to increased sensitivity to pesticides, some common types of which further downregulate the MDR efflux mechanisms in honeybees at sublethal

doses [46]. Understanding the interaction between environmental risk factors and the natural blood-brain barrier functional changes across the honeybee lifespan can inform interventions and management strategies to decrease risk factor impacts on honeybee health [2,7].

We hypothesized that one environmental risk factor that would increase transcellular blood-brain barrier permeability is parasitism by *Varroa*. Our results support this hypothesis, specifically showing honeybees that emerged with 2–4 mites had a significantly higher brain Rho B concentration than those bees that emerged with 0–1 mites (Fig 2B). The reproductive phase of *Varroa* occurs inside of capped brood cells, where the mites bore a hole into the cuticle of the developing brood and feed on its fat body tissue [50]. The depletion of fat body results in organism-wide consequences such as impaired development, reduced lipid synthesis, reduced protein titers, reduced hemolymph sugar levels, and impaired metabolic function [50,51]. At the molecular level, *Varroa* exposure induces transcriptomic and proteomic alterations to immunity, oxidative stress, olfactory recognition, metabolism of sphingolipids, and RNA regulatory mechanisms [52]. Additionally, parasite-induced stress may trigger inflammatory signaling cascades that further compromise blood–brain barrier integrity. In *Drosophila*, neuroinflammatory activation within barrier glia has been shown to disrupt BBB function and increase permeability through immune signaling pathways, suggesting that similar inflammation-related mechanisms could contribute to the elevated transcellular dye penetration observed here [53,54]. We therefore propose that both energetic depletion and inflammation arising from *Varroa* parasitism may synergistically reduce the efficiency of blood-brain barrier efflux transporters, resulting in increased Rho B accumulation in the brains of heavily infested bees.

Across experiments, variability in sample sizes reflects the practical limitations of working with free-living honeybees that develop under natural infection rates. This imbalance, combined with modest group sizes, influenced the statistical sensitivity of our analyses. Power analyses highlight that both aging comparisons were underpowered, with sample sizes sufficient to detect only large effects. For the paracellular assay, the observed medium-sized effect ($f = 0.40$) provided ~36% power, while the transcellular assay ($f = 0.35$) offered similar sensitivity. Consequently, the lack of significant age effects should be interpreted cautiously, as smaller changes in BBB permeability could have gone undetected and may emerge with larger or more balanced groups.

Likewise, the *Varroa* paracellular test achieved only ~6% power, limiting confidence in the null result. In contrast, the *Varroa* transcellular comparison was well powered: with $n = 19$ and 21 bees, the analysis achieved ~83% power to detect the large effect observed (Hedges' $g = 0.95$). Thus, while the *Varroa*-induced increase in transcellular permeability represents a robust finding, the absence of age-related or paracellular effects primarily reflects the limited and unequal sample sizes rather than true biological invariance. The 0–1 mite group was used in these two experiments as a reference condition to represent minimal parasite exposure, with the understanding that bees emerging with one mite may have experienced some stress. Future studies that include a completely mite-free control group will help clarify the specific effects of even minimal *Varroa* exposure on blood–brain barrier function.

A goal of this study was to develop and apply a blood-brain barrier permeability assay that can be widely applicable to different stress contexts. We demonstrated the efficacy of this assay to compare blood-brain barrier permeability in honeybees across two distinct lifespan stages and in honeybees exposed to varying amounts of *Varroa* infestation. While blood-brain barrier permeability is associated with age- and stress-related cognitive dysfunction in multiple species, this study *only* assessed blood-brain barrier integrity. Additional work is required to draw causative connections between blood-brain barrier integrity and honeybee fitness and/or healthspan. A particularly interesting question to probe is how stress is mediated genetically and physiologically at the blood-brain barrier. For instance, a targeted study of blood-brain barrier gene and protein expression under varying amounts of *Varroa* stress will help uncover which barrier efflux proteins are most differentially expressed in response to parasitism. A similar study comparing honeybees at multiple stages of their lifespan can reveal natural ontogenetic changes in blood-brain barrier function, prompting further questions about its role in supporting an aging brain.

 

Although age- and stress-related effects on honeybees can be subtle, it is the accumulation of risk factor impacts which are thought to underlie the devastating colony losses over the past decade.[10] Increased blood-brain barrier permeability is often a sublethal impact, the severity of which can be rapidly assessed with this method. This assay can be particularly useful in pilot studies to identify specific concentrations or combinations of risk factors to focus on for more in-depth analyses of their impact on the honeybee brain across lifespan. It may also be used as a component of monitoring studies for measuring honeybee brain health and overall healthspan over long periods of exposure to anthropogenically-altered environments [55].

## Author contributions

**Conceptualization:** Tyler Quigley, Gro Amdam.

**Formal analysis:** Tyler Quigley.

**Funding acquisition:** Gro Amdam.

**Investigation:** Tyler Quigley.

**Methodology:** Tyler Quigley.

**Project administration:** Gro Amdam.

**Resources:** Gro Amdam.

**Supervision:** Gro Amdam.

**Writing – original draft:** Tyler Quigley.

**Writing – review & editing:** Tyler Quigley, Gro Amdam.

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
