## [Decision Letter · Decision Letter 0]

18 Nov 2024

Dear Dr. Quigley,

Thank you for submitting your manuscript to PLOS ONE. After careful consideration by two reviewers and myself, we feel that it has merit but does not meet PLOS ONE’s publication criteria as it currently stands. Therefore, we invite you to submit a revised version of the manuscript that addresses the points raised during the review process. While I do not necessarily agree with the reviewer comment that additional experiments are necessary at this point (although that would certainly strengthen your manuscript), both reviewers make several really important points that you need to address. In addition, I would like to ask you to perform a statistical power analysis and interpret your results accordingly. Finally, you need to go through your writing thoroughly to eliminate any easily avoided mistakes (e.g., Line 118 or L. 175).

If you are prepared to undertake the required revisions, please submit your revised manuscript by Jan 02 2025 11:59PM. If you will need more time than this to complete your revisions, please reply to this message or contact the journal office at plosone@plos.org . . A rebuttal letter that responds to each point raised by the academic editor and reviewer(s). You should upload this letter as a separate file labeled 'Response to Reviewers'.A marked-up copy of your manuscript that highlights changes made to the original version. You should upload this as a separate file labeled 'Revised Manuscript with Track Changes'.An unmarked version of your revised paper without tracked changes. You should upload this as a separate file labeled 'Manuscript'.

We look forward to receiving your revised manuscript.

Kind regards,

Olav Rueppell

Academic Editor

PLOS ONE

Journal Requirements:

“Research Council of Norway, Grant # 335244”

Reviewers' comments:

Reviewer's Responses to Questions

**Comments to the Author**

1. Is the manuscript technically sound, and do the data support the conclusions?

Reviewer #1: Yes

Reviewer #2: No

2. Has the statistical analysis been performed appropriately and rigorously?

Reviewer #1: No

Reviewer #2: I Don't Know

3. Have the authors made all data underlying the findings in their manuscript fully available?

Reviewer #1: No

Reviewer #2: Yes

4. Is the manuscript presented in an intelligible fashion and written in standard English?

Reviewer #1: Yes

Reviewer #2: No

Reviewer #1: In this manuscript Quigley and Amdam analysed the permeability of the honeybee blood-brain barrier (BBB) during aging and under a parasitic stressor. They found that both paracellular and transcellular permeability are maintained during aging, however, transcellular permeability increases upon exposure to the parasitic mite, Varroa destructor. Overall, this manuscript presents two simple experiments with clear objectives. The experimental design and data presentation are simple and direct, and the conclusions are supported by the data. Nevertheless, some minor adjustments could enhance the manuscript.

In our experience, injecting insects with dyes presents a challenge due to high variability between individuals. Differences in the amount of dye injected in each animal or loss of haemolymph can cause significant variation in tracer concentration. This variability limits the quantitative analysis of the experiments, requiring a large number of animals to obtain a reliable representation of the population.

In this regard, the number of animals is quite variable, ranging from 8 to up 22 honeybees. While limitations in animal availability are understandable, as shown in Table 1, comparing groups of 22 animals to 8 could obscure small differences. The authors should address this issue in the discussion section.

Similarly, in Fig. 1 and Fig. 2, the data are presented as box plots. I strongly suggest that the authors include all points in the plots or provide supplementary tables showing all values. This is particularly relevant, as the authors used parametric statistics but displays median and interquartile ranges as whiskers. Additionally, bimodal distribution may be present in this type of data, making nonparametric statistics more appropriate for analysis.

Minor comments:

Why was a 0-1 mites range used? Would it be more informative to include a completely mite-free (no-stress) control group for comparison?

Additionally, could the authors discuss the possibility of an inflammatory response due to parasite-induced stress? This may be particularly relevant, as studies on Drosophila indicate that inflammation can increase the BBB permeability.

Reviewer #2: Quigley and Amdam report the use of BBB permeability assays in honey bees. These assays have widely been used to study BBB permeability in Drosophila previously. The use of these assays reveals that strong infection with mites increases transcellular BBB permeability in freshly hatched bees. The adaptation of the assays used in Drosophila is surely useful, but rather straight forward given the similarities between the insects. The finding that BBB permeability is changed upon heavy mite infection is interesting, but I have several concerns about the study.

Major concerns:

- It remains unclear if and how many independent experiments have been performed. Have animals from different hives been probed? The number of animals probed is very small and indicates just one independent experiment, which is clearly not enough to draw any general conclusion.

- The authors claim that increased transcellular diffusion over the BBB could have an effect on animal health. However, they do not show if this is really the case. Do bees with increased BBB transcellular permeability show a decrease in fitness? Or at least: Do bees that have been infected by multiple mites show a reduction in health span or fitness?

- The authors show that infection with more than two mites affects transcellular BBB permeability. They probe freshly emerged animals of the age of 1 day. However, it remains unclear if this phenotype persists. The phenotype might be an adaptation to the effect of mite infection during development (reduced resources) and might wear of rapidly after the animal got rid of the mites. In this case, it would probably not impact the animal’s health span or fitness. Thus, I suggest to analyze BBB permeability of animals that have been infected with mites during development at later stages of their life.

- The authors suggest that a follow up study should increase the number of animals tested. I would strongly suggest to do this in this study.

- The authors probe bees of different ages, the oldest age group being 2-3 weeks old, if I am not mistaken. Hoe old does a worker be become usually? Is 2-3 weeks “old” for a bee? I suggest to probe the entire lifespan of a worker to be able to draw conclusions on aging.

- The text (introduction and discussion) describing the features of the BBB contains several mistakes. E.g. the authors state that the BBB is enriched in active transport systems for the exchange of various substances (line 86). There are active transport systems, but many substances are also transported via facilitative transporters (e.g. glucose). Line 158 states that there are tight junctions between the cells – in insects the diffusion barrier is provided by septate junctions. In line 270 the authors claim that there is redundancy between different junctional proteins. Vast research in Drosophila suggests that the junctions are formed by big protein complexed, that comprise many different proteins, which are all needed to keep the complex stable (at least the core proteins). If one of them is missing the complex falls apart. Thus, I would not talk about redundancies when it comes to BBB protein complexes. There are more other imprecisions that should be corrected.

- Several sentences suggest causative connections between unrelated information. Please review the text for inconsistencies and wrong causative relationships. In general, the text does need some proof reading.

**Do you want your identity to be public for this peer review?** For information about this choice, including consent withdrawal, please see our For information about this choice, including consent withdrawal, please see our Privacy Policy .

Reviewer #1: No

Reviewer #2: No

While revising your submission, please upload your figure files to the Preflight Analysis and Conversion Engine (PACE) digital diagnostic tool, https://pacev2.apexcovantage.com/ . PACE helps ensure that figures meet PLOS requirements. To use PACE, you must first register as a user. Registration is free. Then, login and navigate to the UPLOAD tab, where you will find detailed instructions on how to use the tool. If you encounter any issues or have any questions when using PACE, please email PLOS at . PACE helps ensure that figures meet PLOS requirements. To use PACE, you must first register as a user. Registration is free. Then, login and navigate to the UPLOAD tab, where you will find detailed instructions on how to use the tool. If you encounter any issues or have any questions when using PACE, please email PLOS at figures@plos.org . Please note that Supporting Information files do not need this step.. Please note that Supporting Information files do not need this step.

---

## [Author Response · Author response to Decision Letter 1]

18 Dec 2025

We thank the editor for recognizing the merit of our simple yet complete study. We agree that a statistical power analysis and an according interpretation will strengthen the manuscript. We describe our methods (lines 201-218), results (lines 225-240, 286-295), and an interpretation of our results in the context of the power analysis in the Discussion section (lines 416-440).

In this new section, we provide more background on the experimental limitations, namely, our unequal sample sizes, that contribute to the power of our statistical analyses. We explain implications of variation in sample sizes in terms of our statistical analyses. We also draw the readers' attention to results that may have to be interpreted with caution.

---

## [Editor Report · Decision Letter 1]

22 Dec 2025

Dear Dr. Quigley,

plosone@plos.org . . A letter that responds to each point raised by the academic editor and reviewer(s). You should upload this letter as a separate file labeled 'Response to Reviewers'.A marked-up copy of your manuscript that highlights changes made to the original version. You should upload this as a separate file labeled 'Revised Manuscript with Track Changes'.An unmarked version of your revised paper without tracked changes. You should upload this as a separate file labeled 'Manuscript'.

We look forward to receiving your revised manuscript.

Kind regards,

Olav Rueppell

Academic Editor

PLOS One
---

## [Author Response · Author response to Decision Letter 2]

28 Jan 2026

Dear Dr. Rueppell,

Thank you for the opportunity to resubmit a revised draft of the manuscript titled “Effect of aging and Varroa parasitism on the paracellular and transcellular permeability of the honeybee blood-brain barrier” for publication in PLOS ONE. We greatly appreciate the time and effort that you have committed to reviewing and providing feedback on our manuscript. We have incorporated your additional suggestions. The changes we’ve made are in track changes in the manuscript, and have addressed each piece of feedback directly below in red.

Editor Comments:

1. Simple check of spelling and grammar throughout (e.g., in the abstract, the statement "we assess a key dimensions of blood-brain barrier function")

We thank the editor for noticing this discrepancy and have review the entire manuscript to ensure that all other spelling, grammar, missing word, and other minor clarifying edits are fixed (lines 33, 34, 44, 60, 92, 96, 103, 108, 139, 152, 160, 164, 171, 180, 187, 190, 192, 204, 206, 210, 211, 223, 226, 231, 235, 250, 252, 253, 260, 261, 266, 267, 273, 274, 281, 289, 290, 296, 297, 306, 325, 329, 336, 341, 352, 364, 400, 410, 414)

2. The issue about your way of testing age or ageing: It needs to be clear in the abstract and discussion that you had limited control over the chronological age of workers bees, and that you are comparing age ranges that experienced varying amounts of foraging (although of course your groups are different on average). Whether you really have included "old workers" is not certain in the absence of better demographic data on your study subjects.

We thank the editor for pointing out this necessary clarification. We have made edits to the abstract (lines 35-36, 39, 41-42) and added two sentences to the discussion (lines 290-293, 337-340) that highlight this specific limitation in our sampling strategy.

3. The significance levels in figure 2 seem to be switched, indicating 2A as significant and 2B as N.S.

We thank the editor for pointing this out, we have made the relevant correction to Figure 2 (line 271).

---

## [Editor Report · Decision Letter 2]

3 Feb 2026

Effect of aging and Varroa parasitism on the paracellular and transcellular permeability of the honeybee blood-brain barrier

PONE-D-24-43745R2

Dear Dr. Quigley,

We’re pleased to inform you that your manuscript has been judged scientifically suitable for publication and will be formally accepted for publication once it meets all outstanding technical requirements.

Kind regards,

Olav Rueppell

Academic Editor

PLOS One
---

## [Editor Report · Acceptance letter]

PONE-D-24-43745R2

PLOS One

Dear Dr. Quigley,

I'm pleased to inform you that your manuscript has been deemed suitable for publication in PLOS One. Congratulations! Your manuscript is now being handed over to our production team.

Kind regards,

on behalf of

Dr. Olav Rueppell

Academic Editor

PLOS One